# microRNA-mRNA Analysis Reveals Tissue-Specific Regulation of microRNA in Mangrove Clam (*Geloina erosa*)

**DOI:** 10.3390/biology12121510

**Published:** 2023-12-11

**Authors:** Yunqing Liu, Ziheng Dong, Kun Chen, Mingliu Yang, Nianfeng Shi, Xin Liao

**Affiliations:** 1School of Computer and Information Engineering, Luoyang Institute of Science and Technology, Luoyang 471023, China; liuyq@lit.edu.cn (Y.L.); zihengdong0@gmail.com (Z.D.); 2Guangxi Key Laboratory of Mangrove Conservation and Utilization, Guangxi Academy of Marine Science (Guangxi Mangrove Research Center), Guangxi Academy of Science, Beihai 536007, China

**Keywords:** *Geloina erosa*, miRNA, tissue-specific expression, co-expression network analysis, target gene prediction, immune response, metabolism, environmental adaptation

## Abstract

**Simple Summary:**

*Geloina erosa*, an important animal in mangrove ecosystems, possesses both economic and environmental value. However, to date, there has been no research conducted on its miRNA. This study utilized high-throughput sequencing to identify 1412 miRNAs in three tissues: the gills, hepatopancreas, and muscles. Among these, 1047 miRNAs were identified as known conserved sequences. In conjunction with mRNA research, it was observed that approximately 33% of the transcripts were predicted to be predicted targets for miRNAs. Building upon our team’s previous research on mRNA, it is worth noting that the majority of genes suppressed by miRNAs in different tissues align with their respective organ functions. Specifically, in gills, miRNAs predominantly regulated immune-related genes, substance transport, and cytoskeletal organization. In the hepatopancreas, miRNAs suppress genes involved in shell formation while also influencing cellular motor activity and metabolism. In muscle tissue, miRNAs were found to participate in metabolism, photoreceptive processes, and immune regulation. These findings suggest that miRNA regulation is finely tuned to swiftly respond to environmental changes. Overall, these discoveries provide significant insights into the molecular mechanisms and biological processes of miRNA within *G. erosa*. They serve as a foundation for further validation and elucidation of these regulatory relationships.

**Abstract:**

*Geloina erosa* is an important benthic animal in the mangrove, serving as an indicator organism for coastal environmental pollution. This study aimed to investigate the tissue-specific expression of miRNAs and their regulatory roles in predicted targets in *G. erosa*. Through miRNA sequencing and co-expression network analysis, we extensively studied the miRNA expression in three tissues: gills, hepatopancreas, and muscle. The results revealed a total of 1412 miRNAs, comprising 1047 known miRNAs, and 365 newly predicted miRNAs. These miRNAs exhibited distinct tissue-specific expression patterns. In the miRNA target gene prediction, a total of 7404 potential predicted targets were identified, representing approximately 33% of all unique transcripts associated with miRNAs. Further co-expression network analysis revealed nine modules, each showing a positive correlation with specific tissues (gills, hepatopancreas, or muscle). The blue module showed a significant correlation with gills (r = 0.83, *p*-value = 0.006), the black module was significantly related to the hepatopancreas (r = 0.78, *p*-value = 0.01), and the purple module was significantly correlated with muscle (r = 0.83, *p*-value = 0.006). Within these modules, related miRNAs tended to cluster together, while their correlations with other modules were relatively weak. Functional enrichment analysis was performed on miRNAs and their predicted targets in each tissue. In the gills, miRNAs primarily regulate immune-related genes, substance transport, and cytoskeletal organization. In the hepatopancreas, miRNAs suppressed genes involved in shell formation and played a role in cellular motor activity and metabolism. In muscle, miRNAs participate in metabolism and photoreceptive processes, as well as immune regulation. In summary, this study provides valuable insights into the tissue-specific regulation of miRNAs in *G. erosa*, highlighting their potential roles in immune response, metabolism, and environmental adaptation. These findings offer important clues for understanding the molecular mechanisms and biological processes in *G. erosa*, laying the foundation for further validation and elucidation of these regulatory relationships.

## 1. Introduction

miRNAs are widely present in organisms, including animals, plants [1], and bacteria [2]. The mechanism of miRNA action in mammals has been well studied [3], while in invertebrates it remains limited and requires further investigation [4]. Umberto Rosani et al. [5] summarized the biogenesis-related genes of miRNAs in bivalves and found that the synthesis mechanisms of miRNA are conservative. In recent years, more and more studies have focused on the regulatory mechanisms of miRNAs in bivalves. In bivalves, the predicted targets regulated by miRNAs are widely involved in various biological functions [6]. In the freshwater pearl mussel, miR-9 participates in sex differentiation by regulating their predicted targets *OXL2* and *STAR1* [7,8,9]. In *Perna viridis*, miRNAs are involved in detoxification. In *Patinopecten yessoensis*, miRNAs play an important regulatory role in fast (striped) and slow (smooth) muscle fiber formation [10]. miRNAs also play a role in the formation of bivalve shells in special tissue [11,12,13]. Studies on mussels have found significant tissue and condition-specific expression patterns of miRNAs, which mean that miRNAs play an important role in the regulation of tissue-specific functions. [14]. Given the pivotal roles that miRNAs play in bivalves, understanding their expression patterns in specific species can offer crucial insights into their biology and adaptation mechanisms. In this context, our research focuses on *G. erosa*, using it as a material to explore the miRNA expression patterns across different tissues.

*G. erosa* holds substantial economic value and is targeted by coastal fishermen in some area for its large size of up to 10 cm [15,16]. *G. erosa* primarily inhabits estuarine zones and serves as an important component of mangrove ecosystems [17,18]. Additionally, given its wide distribution and abundance, it can serve as an indicator organism for coastal environmental pollution. However, research on its gene and cellular aspects remains relatively limited.

Conducting transcriptome expression analysis under environmental stress or pollution conditions [19,20] has become a major approach in this field. Therefore, the study of gene expression under physiological conditions has important reference values. Previously, we focused on analyzing the tissue-specific gene expression characteristics at the transcriptome level and discovered co-expression regulatory networks among genes. In this study, we focus on miRNA research in *G. erosa* by elucidating the tissue-specific expression patterns of miRNAs and their impact on gene regulation. Specifically, we aim to answer three key research questions: First, what distinct miRNA expression profiles can be observed in the gills, hepatopancreas, and muscles? Second, which predicted targets are specifically regulated by these miRNAs in each tissue? And third, do these miRNAs and their predicted targets contribute to the unique physiological functions of each tissue? To address these questions, we chose to examine the gills, hepatopancreas, and muscles—tissues commonly studied in *G. erosa* research. Among these tissues, the hepatopancreas serves as the primary digestive and detoxification tissue, while the gills are primarily responsible for substance exchange with the external environment. Muscles were used as control samples to better understand the specific roles of miRNAs in gill and hepatopancreas functions [21].

## 2. Materials and Methods

### 2.1. Collection and Preparation of Animal Material

The animal samples for sequencing were collected from mangrove wetlands located in Beihai, Guangxi, China (21.57′ N, 109.16′ E). No specific permissions were required for sample collection from these locations, and the field studies did not involve any endangered or protected species. The welfare of the animals and experimental procedures adhered to the ethical regulations of the Guangxi Mangrove Research Center and followed the guidelines outlined in the Guide for the Care and Use of Laboratory Animals. We collected nine samples, with S1–S3 representing gills, S4–S6 representing the hepatopancreas, and S7–S9 representing muscle (Table 1).

### 2.2. miRNA Extraction and Quality Control

Nine samples, comprising three tissues (hepatopancreas, gill, and muscle), each with three biological replicates, underwent rapid separation on ice to isolate the total RNA. The extraction of total RNA was performed employing the Tiangen RNAprep Pure Tissue kit, manufactured by Tiangen Biotech in Beijing, China.

The RNA samples underwent multiple quality control measures to ensure their suitability for downstream applications. Firstly, the purity of the samples was assessed using Nanodrop measurement, with criteria set at an OD260/280 ratio of ≥1.8 and an OD260/230 ratio of ≥1.0. Secondly, the Qubit 2.0 measurement accurately quantified the concentration of total RNA, with a minimum requirement of ≥250 ng/µL. Finally, the RNA samples were subjected to Agilent 2100 bioanalyzer analysis, including evaluation of RNA integrity through parameters such as a total RNA RIN value of ≥8.0, 28 S/18 S ratio of ≥1.5, a flat baseline spectrum, and the presence of a normal 5 S peak. These comprehensive assessments ensured the usage of qualified RNA samples for subsequent sequencing applications.

### 2.3. Library Preparation, Library Quality Control, and Sequencing

After passing the qualification tests, the samples underwent initial RNA extraction, resulting in 1.5 µg of RNA. The solution volume was then adjusted to 6 µL using DEPC-treated water. Library construction was carried out strictly following the instructions of NEB Next Ultra small RNA Sample Library Prep Kit for Illumina. Small RNAs have a phosphate group at their 5′ end and a hydroxyl group at their 3′ end. Therefore, T4 RNA Ligase 1 and T4 RNA Ligase 2 (truncated) were respectively employed to ligate adapters to the 3′ and 5′ ends of the small RNAs. Subsequently, reverse transcription was performed to synthesize cDNA, followed by PCR amplification. Gel electrophoresis facilitated the selection of desired fragments, constituting the small RNA library. The workflow for library construction is presented in Figure 1.

The small RNA libraries were sequenced using the Illumina HiSeq2500 platform. Subsequently, the raw image data files obtained from sequencing underwent base calling, leading to the generation of raw sequencing sequences, also known as raw data or raw reads. These outcomes were then stored in the FASTQ file format, which includes both the sequence information of the sequencing reads and their corresponding sequencing quality data.

### 2.4. Bioinformatics Analysis

#### 2.4.1. Overview of Bioinformatics Analysis

The small RNA analysis workflow begins with the preprocessing of the raw sequencing data, followed by quality control to ensure the sequencing data’s reliability. This step includes assessing the length distribution and analyzing the common and unique sequences.

After quality control, the small RNA sequences undergo classification and annotation. Various categories are identified, such as snoRNA, scRNA, repeat sequences, rRNA, tRNA, and snRNA. This process helps in distinguishing known miRNAs and predicting new miRNAs.

Once the miRNAs are identified, further analysis can be conducted, which includes studying the miRNA structure, exploring miRNA families, examining potential nucleotide edits, and predicting predicted targets for the miRNAs. Additionally, the preference for specific nucleotides in the miRNA sequences can be analyzed.

Following these steps, differential expression analysis is performed to identify miRNAs that exhibit significant changes in expression levels between different tissues. This analysis provides insights into miRNA regulation under specific conditions.

Finally, the differentially expressed miRNAs are subjected to functional annotation of their predicted targets, providing information on the potential biological functions associated with the differential expression. Additionally, enrichment analysis is performed to identify functional categories or pathways that are overrepresented among the predicted targets of differentially expressed miRNAs. This analysis helps in understanding the functional implications of the identified miRNAs.

#### 2.4.2. Quality Control of Sequencing Data

The raw miRNA sequencing data underwent quality control, involving the removal of low-quality sequences, trimming of adapter sequences, filtering of low-quality bases, and other relevant procedures, to ensure the reliability of subsequent analyses.

In this study, sequencing quality was evaluated using metrics based on sequencing base quality and data output statistics. The base quality, represented as a quality score or Q-score, quantifies the probability of base-calling errors. The widely used Phred formula for calculating base quality scores is Q-score = −10 × log_10_P, where P represents the probability of base-calling error [22].

#### 2.4.3. Identification of Novel miRNAs and Homologous miRNAs

Known miRNA identification involved comparing unannotated read sequences with mature miRNA sequences from the known miRNA database miRBase (v21). Reads having the exact same sequence as known miRNAs were classified as identified known miRNAs.

The miRNA transcription initiation sites are mostly located on the gene spacer region, intron, and reverse complementary sequence of the coding sequence, their precursors have a signature hairpin structure, and the formation of mature bodies is realized by the cleaving of Dicer/DCL enzymes. For the biological characteristics of miRNAs, miRDeep2 software (v2.0.5) [23] was used to predict new miRNAs for sequences that were not similar to the known miRNAs in the database through sequence comparison.

We used the miRDeep2 software package to obtain possible precursor sequences by comparing reads to the position information on the genome, and based on the distribution information of reads on the precursor sequence (based on miRNA production characteristics, mature, star, loop) and precursor structural energy information (RNAfold randfold), the prediction of new miRNAs was finally achieved by scoring using a Bayesian model. miRDeep2 is mainly used for the prediction of animal miRNAs, and the miRNA of plants can also be predicted by adjusting parameters and changing the scoring system [24].

#### 2.4.4. miRNA Target Gene Prediction

miRNAs typically play a regulatory role by inhibiting the expression of predicted targets. Based on the genetic sequence information of known miRNAs and newly predicted miRNAs and corresponding species, MiRanda [25] and RNAhybrid [26] for animal target gene prediction was performed.

#### 2.4.5. miRNA Expression Quantification

The expression of miRNA in each sample was counted via expectation maximization (RSEM) [27] and the TPMs (transcripts per million) algorithm was used. The expressions were normalized [28].

#### 2.4.6. Differential Expression Analysis

When detecting differentially expressed miRNAs, it is necessary to select the appropriate differential expression analysis software according to the actual situation. DESeq2 (v1.18.0) [29] is suitable for experiments with biological replicates, and the differential expression analysis between sample groups can be performed to obtain the difference expression miRNA set between the two biological conditions.

In the detection of differentially expressed miRNAs, |log_2_(FC)| ≥ 1 was used; a FDR (false discovery rate) ≤ 0.05 was the screening criterion. The fold change (FC) represents the ratio of expression between two samples (groups). The significance *p*-value obtained by the original hypothesis can be expressed as the probability of expressing no difference. Since the differential expression analysis of miRNA is an independent statistical hypothesis test for a large number of miRNA expressions, there is a false positive problem, so in the analysis process, the Benjamini–Hochberg correction method is sometimes used to correct the significance *p*-value obtained by the original hypothesis test, and finally, the FDR is used as the key indicator of differential expression miRNA screening.

The set of miRNAs found via differential expression analysis is called the differential expression miRNAs (DEMs).

#### 2.4.7. Functional Annotation of Predicted Targets

The gene functions were annotated using the following databases: NCBI non-redundant protein sequences (NR) [30], protein family (Pfam) [31], Clusters of Orthologous Groups of proteins (KOG/COG) [32], Swiss-Prot [33], Kyoto Encyclopedia of Genes and Genomes Ortholog database (KEGG) [34], and Gene Ontology (GO) [35]. We used Diamond BLASTX software (v2.2.26) and an e-value of “1 × 10^−10^” when performing NR, KOG, Swiss-Prot, and KEGG database analysis. Hmmscan was used for family identification with the Pfam database.

#### 2.4.8. Gene Set Enrichment Analysis

In this study, the researchers employed the Molecular Signatures DataBase (MSigDB) software (v4.3.2), developed by [36], to assess the statistical enrichment of genes. To categorize transcripts into different gene classes, databases such as KEGG, GO, KOG, and Pfam were utilized.

To determine the enrichment of each gene class, Fisher’s exact test was conducted either on the ordered genes or a gene list specified by the user. A significance threshold of *p* < 0.05 was considered to identify gene classes with significant enrichment.

This comprehensive analysis allowed valuable insights to be gained into the functional characteristics and associations of the identified genes. By elucidating their potential roles and involvement in various biological processes, the findings provide important information regarding their significance.

The utilization of the MSigDB software (v4.3.2) and the incorporation of diverse gene classification databases enhanced the understanding of gene enrichment, enabling the uncovering of meaningful relationships and shedding light on the functional implications of the genes under investigation.

#### 2.4.9. miRNA Co-Expression Analysis

The widely used weighted gene co-expression network analysis (WGCNA) [37] was employed in this study as a powerful method to identify co-expressed miRNAs and hub miRNAs. It has proven instrumental in identifying co-expressed gene modules and establishing associations between these modules and traits [38]. Using miRNA expression data, the WGCNA software (v 1.72.1) was utilized to analyze associations between modules and tissues or traits of interest. Within undirected networks, miRNAs within the same module exhibit high levels of interrelatedness. After clustering miRNAs into modules, two key analyses were conducted as follows:(1)A functional enrichment analysis was employed to determine whether the functional characteristics and traits identified within each module aligning with the specific aims of the research. This analysis helped gain insights into the biological processes and pathways that are potentially associated with the study objectives.(2)A correlation analysis was performed to identify the module that exhibited the highest correlation with the traits of interest. By exploring the relationship between miRNA expression patterns within modules and specific traits, the aim was to uncover modules that potentially play crucial roles in the biological processes underlying these traits.

To ensure robust results, the parameters were set as follows for all samples: the power was set to 12, minModuleSize was set to 30, and maxBlockSize was set to the total number of miRNAs in the analysis. These parameter settings were chosen to optimize the identification of meaningful miRNA modules and trait associations within the dataset.

#### 2.4.10. Real-Time Fluorescent Quantitative PCR Validation of miRNA Expression in Tissues

Six miRNAs were randomly selected for the validation of sequencing results through real-time fluorescence quantitative PCR (qRT-PCR). The validation experiment used clams collected from the Dongwei Mangrove in Beihai City (21.55° N, 109.16° E). Three types of tissues, muscle, gills, and hepatopancreas, were collected. Each tissue was taken from 9 individual samples, with 3 biological replicates set up. Every three tissue samples were mixed for RNA extraction. The total RNA from each tissue was extracted using the TaKaRa MiniBEST Universal RNA Extraction Kit, following the instructions provided in the kit manual. The purity of the extracted RNA was assessed using a NanoDrop spectrophotometer, while RNA integrity was verified by agarose gel electrophoresis. Primers were designed using the Primer premier 5.0 software, as shown in Table 2. The Takara Mir-X miRNA FirstStrand Synthesis and TB Green qRT-PCR Kit were used for the reverse transcription of total RNA into cDNA and subsequent qRT-PCR experiments. The primers for the internal reference snRNA U6 and the universal miRNA reverse primer were provided in the kit. The miRNA expression relative quantification experiments and analyses were carried out on the QuantStudio 3 Real-Time PCR System. The PCR reaction volume was set at 20 µL, and the PCR protocol was followed as per the product instructions. Each sample underwent 3 biological repeats and 3 technical repeats. The relative miRNA expression was calculated using the 2^−ΔΔCt^ method. Data processing and plotting were performed using GraphPad 8.0.

## 3. Results

### 3.1. Sequencing Results and Quality Statistics

In this study, the number of clean reads obtained for each sample ranged from 25 million to 35 million. The Q30 is above 97%, indicating high sequencing quality (Table 3).

It can be concluded from Figure 2 that the sequencing quality and consistency of the samples are good, conforming to the general distribution of miRNA sequencing data.

### 3.2. miRNA Identification

After analysis, a total of 1412 miRNAs were obtained from all samples, including 1047 known miRNAs and 365 newly predicted miRNAs (Table 4).

Due to the specificity of the Dicer enzyme and DCL enzyme, the length of mature miRNA finally generated was mainly concentrated in the range of 20 nt to 24 nt. The length distribution of the identified new miRNAs is shown in Figure 3C.

### 3.3. miRNA Base Preference Analysis

The Dicer enzyme and DCL enzyme recognize and cleave the precursor miRNA with a strong bias towards U at the 5′ end. The typical miRNA base ratios were obtained via a base preference analysis of miRNA. The statistics of the first base preference at the 5′ end of the newly predicted miRNA and the base preference of each point are shown in Figure 3A,B.

### 3.4. miRNA Target Gene Prediction

In this study, among the 1047 known miRNAs, 700 have corresponding predicted targets, resulting in a total of 5082 predicted targets. Additionally, among the 365 newly predicted miRNAs, 334 were found to have corresponding predicted targets, accounting for 3223 targets (Table 5). The total number of predicted targets for all miRNAs was 7404, which represents approximately 33% of all unique transcripts. This finding highlights miRNA as one of the primary methods for gene expression regulation in *G. erosa* (Table 5). Across the three tissues analyzed, the number of miRNAs ranged from 800 to 1200. Notably, the highest number of miRNAs was expressed in the gills, with a relatively large number of corresponding predicted targets. Conversely, the lowest number of miRNAs and their corresponding predicted targets was observed in muscle (Table 6).

### 3.5. miRNA Expression Quantification

#### 3.5.1. Distribution of Overall Expression in the Sample

The overall distribution of miRNA expression reflects the overall expression pattern of miRNA in the sample, as shown in Figure 4.

#### 3.5.2. Differential Expression Analysis

The differential expression of miRNAs between different tissues was compared in this study. One of the three tissues served as the case group, while the remaining two tissues were used as the control group for separate differential expression analyses. In comparison with the control group, we observed the following differential expressions of miRNAs: in the gill tissue, 96 miRNAs were up-regulated and 112 miRNAs were down-regulated; in the hepatopancreas tissue, 61 miRNAs were up-regulated and 97 miRNAs were down-regulated; in the muscle tissue, 100 miRNAs were up-regulated and 119 miRNAs were down-regulated. All differential expression results are detailed in Appendix A (Figure 5).

#### 3.5.3. Functional Analysis of Differentially Expressed miRNA Predicted Targets

To study the tissue-specific miRNA regulatory network, we overexpressed up-regulated miRNAs in specific tissues (gills, hepatopancreas, and muscle) while simultaneously down-regulating the expression of their corresponding predicted targets. The research objectives focused on cases where the FDR (false positive discovery rate) of the two was less than 0.05. We further conducted a detailed analysis of the molecular functions and enrichment results of key predicted targets.

For the investigation of predicted target expression and regulation by miRNAs in each tissue, we identified miRNAs that are significantly upregulated (FDR < 0.05, log fold change > 1) in a specific tissue, while their corresponding predicted targets that are significantly down-regulated (FDR < 0.05, log fold change < −1) in that tissue were identified as potential miRNA–mRNA pairs with regulatory relationships. These sets of candidate miRNA-regulated predicted targets underwent enrichment analysis (Appendix A). We found 10 gene sets were significantly enriched in the hepatopancreas (Figure 6, Appendix A), mainly involved in protein processing and transport, as well as stress response. Additionally, 26 gene sets significantly enriched in the gills (Figure 7, Appendix A), primarily related to energy metabolism, cytoskeleton, and material transport functions. In the muscles, 16 gene sets showed significant enrichment (Figure 6, Appendix A), primarily related to cell motility and metabolism.

Among all the predicted miRNA-regulated predicted targets, the gills exhibited the highest number of predicted targets and the most enriched related functions. While the hepatopancreas had more predicted targets than muscle, there might be fewer actual regulatory relationships under physiological conditions. We speculate that hepatopancreas-related regulatory mechanisms may be activated under specific conditions, such as stress, which requires further experimental verification.

#### 3.5.4. miRNA Co-Expression Analysis

In the co-expression network analysis, we obtained a total of nine modules. Taking the three tissues as the three phenotypes, a correlation analysis with the nine modules was performed, and it was found from the results that each module was positively correlated with one phenotype but had nothing to do with or was negatively correlated with other modules. The templates positively correlated with the gills include magenta, pink, and blue, and the blue module is significantly correlated with the gills (r = 0.83, *p* value = 0.006). Modules significantly related to the hepatopancreas include red, black, green, and yellow, and the black module is significantly related to the hepatopancreas (r = 0.78, *p* value = 0.01). The modules that are significantly correlated with purple include purple and gray, and the purple module is significantly correlated with muscle (r = 0.83, *p* value = 0.006) (Figure 8A). Module clustering revealed that modules related to specific tissues generally clustered together (Figure 8B). The miRNA clustering heatmap (Figure 8C) conveys high correlation within the miRNA module, and low correlation between modules (Appendix A). The predicted target genes enrichment of miRNAs related to the modules are shown in Figure 7B and Figure 9.

#### 3.5.5. qRT-PCR and RNA-seq Consistency for Selected miRNAs

The qPCR relative expression levels of six miRNAs in the three tissues and the TPM from RNA-seq results are shown in Figure 10. All six miRNAs showed consistent qRT-PCR expression patterns as the high throughput sequencing data. The results show that the results of the RNA-seq technique are credible.

## 4. Discussion

Tissue-specific functions result from the selective expression of genes, which are regulated at both the transcriptional and post-transcriptional levels. While the former represents the primary mode of regulation in eukaryotes and mainly determines whether tissue-specific genes are expressed, the latter offers a more refined level of control, further modulating the expression levels of tissue-specific genes. In this context, miRNAs serve as a part of the post-transcriptional regulation. Highly expressed miRNAs in tissues often target tissue-specific genes. In mammals, for example, miR-206 is highly expressed in muscle tissue, and plays a crucial role in muscle development and regeneration by regulating genes involved in muscle differentiation and function [39]. Similarly, miR-122, abundant in liver tissue, regulates genes involved in hepatic metabolism, cholesterol homeostasis, and viral replication [40]. In pancreatic islet tissues, miR-375 regulates insulin secretion and pancreatic beta-cell function by targeting genes involved in glucose metabolism and insulin signaling [41]. The findings of this study are consistent with the above literature.

### 4.1. Gill-Specific miRNA and Target Gene Functions

One function of the gills is defending against the invasion of external pathogens. They are one of the most important immune barriers in the *G. erosa*. In our study on the regulatory relationships in gill-specific regulation, we identified the target gene F01.PB12493 regulated by the miRNA str-miR-34c-5p, which possesses urease activity and is mainly involved in the process of arginine metabolism. This process plays various important roles, including immunity, in multiple organisms. Studies have shown that inhibiting the metabolism and transport of arginine significantly reduces pathogen invasion and enhances the body’s defense capabilities [42]. The target gene F01.PB21580 regulated by the miRNA sme-miR-31b-5p belongs to the thiol ester-containing protein (TEP) B family. The TEP protein family is highly conserved among species and plays an important role in innate immunity [43]. Studies in oysters have shown high expression of this gene in multiple tissues, especially in the gills. However, our study revealed significant suppression of its expression. Another gene of the TEP protein family, F01.PB12484, is annotated as CD109 antigen and is suppressed by dan-miR-92b [44]. Additionally, F01.PB11852, regulated by age-miR-34a, is involved in the biological defense response process (GO annotation). A KEGG pathway enrichment analysis revealed significant enrichment in the immune-related phagosome pathway. In the co-expression analysis, the blue module showed significant correlation with the gills, and the corresponding predicted targets were enriched in 87 gene sets. They mainly involve metabolic processes, such as gluconeogenesis, the glycolytic process, fatty acid metabolic process, and cellular lipid metabolic process. Related processes involved in calcium ion regulation include cellular calcium ion homeostasis and calcium ion transmembrane transport.

Another important function of the gills is filter feeding, which is an energy-consuming process that requires the participation of energy metabolism and involves material transport. Among the 26 downregulated predicted targets regulated by significantly upregulated miRNAs in the gills, the GO annotations include the GTP catabolic process, ATP catabolic process, and other energy metabolism processes. The subcellular localization is in the cytoplasm, and the corresponding molecular function is ATPase activity. Under conditions of stress, such as hypoxia, bivalves adapt to the environment by rapidly adjusting their metabolic processes [45]. In aquatic invertebrates, the regulation of intracellular osmotic pressure can be achieved through the transport of amino acids and other substances [46]. The enriched KEGG pathways mainly relate to the metabolism of carbohydrates, lipids, and amino acids, as well as the FoxO signaling pathway. In bivalves, processes such as gluconeogenesis [47,48] and glycosphingolipid biosynthesis [49] are associated with resistance to pathogen invasion.

### 4.2. Hepatopancreatic-Specific miRNA and Target Gene Functions

The hepatopancreas is the main site of metabolism and detoxification in the *G. erosa*. In the co-expression analysis, the black module showed a significant correlation with the hepatopancreas. This module enriched a total of 49 gene sets. The corresponding GO functions include the regulation of the cytoskeleton, such as the structural constituent of the cytoskeleton; regulation of metabolic processes, such as acyl-CoA dehydrogenase activity and malate dehydrogenase; transmembrane transport of substances; and negative regulation of the apoptotic process. In a study on the razor clam *Sinonovacula constricta*, it was observed that anti-apoptosis-related genes are upregulated in response to bacterial and other pathogen invasions [50]. The KEGG pathway analysis revealed enrichment in the “Other glycan degradation” pathway, and a previous study discovered HS-like glycosaminoglycan, suggesting its potential role in rapid blood sugar regulation through glycogen [51].

The metabolic process of the liver also involves a large number of substance transport processes. The gene F01.PB20600, regulated by miRNA unconservative F01.PB18584 393223, interacts with kinesin and participates in microtubule-based processes. A study on oysters showed that miRNA-regulated biological processes include microtubule formation and cellular component movement, which are significantly upregulated under environmental stress, suggesting their involvement in altered cellular metabolism processes [52]. In the hepatopancreas, a small number of genes (10 genes) are significantly upregulated by miRNAs. These genes are mainly associated with intracellular substance transport activities such as vesicle-mediated transport, intracellular protein transport, and response to stress activities such as overexpression of the Hsp70 protein family. In bivalves, which have digestive gland secretory cells, vesicle-mediated transport is the main pathway for secretion processes [53].

This study also found that genes involved in shell formation are significantly suppressed in the hepatopancreas. The gene F01.PB11180, regulated by miRNA unconservative_F01.PB16964_362242, is annotated as chondroitin 4-sulfotransferase 11, which plays a crucial role in biomineralization processes. However, in the hepatopancreas, biomineralization is inhibited by the suppression of this gene through miRNA regulation. Another gene, F01.PB14416, regulated by miRNA unconservative_F01.PB18659_394923 is associated with calcium ion binding and is essential for the construction and maintenance of shells in bivalve animals [54].

### 4.3. Muscle-Specific miRNA and Target Gene Functions

Significantly upregulated miRNA-regulated predicted targets were found to be enriched in cellular motor activity and metabolism in muscle. These genes were involved in processes such as transferring glycosyl groups, actin binding, and calmodulin binding. The movement process of muscles requires a large amount of energy support. The gene F01.PB18339, regulated by miRNA efu-miR-133-3p, was identified as a ubiquitin-protein ligase. It is speculated that this gene may be involved in protein degradation inhibition or extracellular matrix protein conversion [55]. In co-expression analysis, the purple module was significantly associated with muscle and enriched with 42 gene sets. The related biological processes included regulation of the cell cycle and regulation of macromolecule metabolic process. The enriched KEGG pathway was amino sugar and nucleotide sugar metabolism. In both GO and KEGG, gene sets related to oxidative stress metabolism were enriched, particularly in PFAM, corresponding to the structure of Iron/manganese superoxide dismutases, C-terminal domain.

Another function involved in the high expression of miRNA in muscles is photoreceptive processes. In a previous study [56], it was found that genes associated with photoreception were highly expressed in muscle tissue. In this study, the gene F01.PB22530, regulated by miRNA gga-miR-133c-3p, exhibited retinal dehydrogenase activity.

In muscle tissue, miRNAs were also observed to participate in immune regulation. The gene F01.PB3499, regulated by miRNA sme-miR-133a-3p, exhibited serine-type peptidase activity (GO:0008236). The gene F01.PB13824, regulated by miRNA unconservative_F01.PB13824_261438 is a CD109 antigen (A) gene.

In both the gills and muscle, miRNAs regulate the FoxO signaling pathway, which plays a crucial role in individual development, cell proliferation, stress response, and metabolism in bivalves [57,58].

## 5. Conclusions

This study provides the first overview of the expression of miRNAs in the *G. erosa*. Our findings indicate that: 1. The predicted targets regulated by miRNAs are tissue-specific or marker genes associated with the expression of those miRNAs, indicating their significant role in the fine-tuned regulation of tissue functions. 2. Approximately one-third of the genes in the *G. erosa* were predicted to be predicted targets of miRNAs, highlighting the importance of miRNA-mediated post-transcriptional regulation in this species. 3. Clustering and co-expression analysis revealed significant tissue-specific expression patterns of miRNAs. Immune-related and metabolic miRNAs were widely expressed in all three tissues, while genes involved in light sensitivity regulation were expressed and regulated by miRNAs in muscle tissue. Overall, this study enhances our understanding of miRNA-mediated regulatory networks in the *G. erosa* and provides a foundation for further research. It should be pointed out that the interaction between miRNA and genes in vivo requires further experimental verification.

## Figures and Tables

**Figure 1 biology-12-01510-f001:**
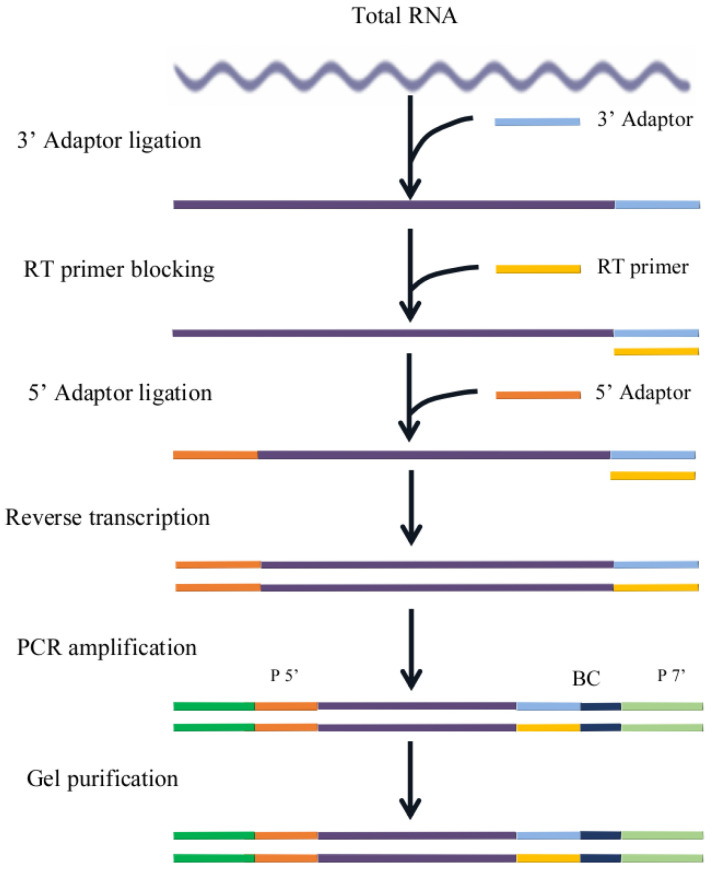
The workflow for library construction.

**Figure 2 biology-12-01510-f002:**
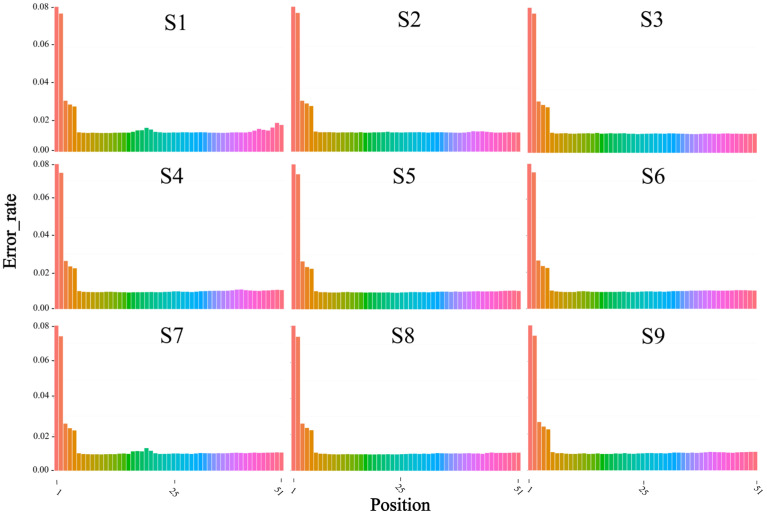
Base quality distribution map. The abscissa is the base position of the reads, and the ordinate is the average value of the base error rate of each sequencing reaction. S1–S9 represent 9 samples, respectively.

**Figure 3 biology-12-01510-f003:**
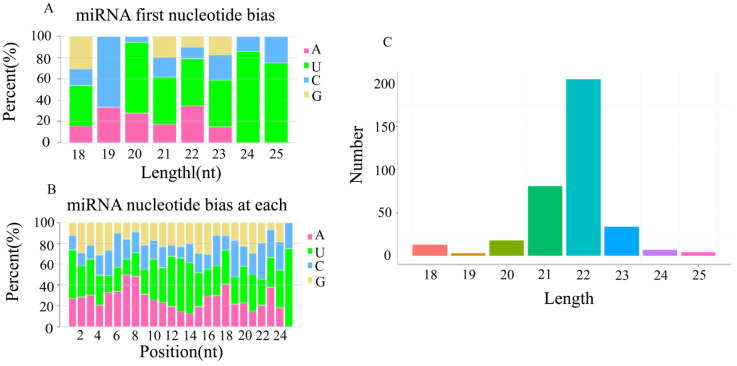
miRNA base preference. (**A**) The first base distribution of miRNAs of different lengths. The abscissa indicates sequences of different lengths; the ordinate indicates the percentage of the first base of each length of miRNA. (**B**) miRNA site base distribution. (**C**) Length distribution of new predicted miRNAs.

**Figure 4 biology-12-01510-f004:**
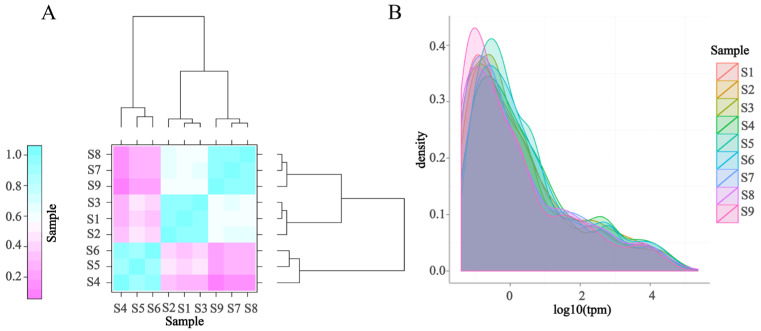
Sample population expression. (**A**) Sample correlation plot. The different colors in the plot represent different correlation coefficient values. The horizontal and vertical coordinates represent different samples. (**B**) Density distribution of sample expression values. The different-colored curves in the TPM density distribution chart represent different samples, the abscissa of the points on the curve represents the logarithmic value of the corresponding sample TPM, and the ordinate of the points represents the probability density. S1–S3 represents gills, S4–S6 represents hepatopancreas, and S7–S9 represents muscle.

**Figure 5 biology-12-01510-f005:**
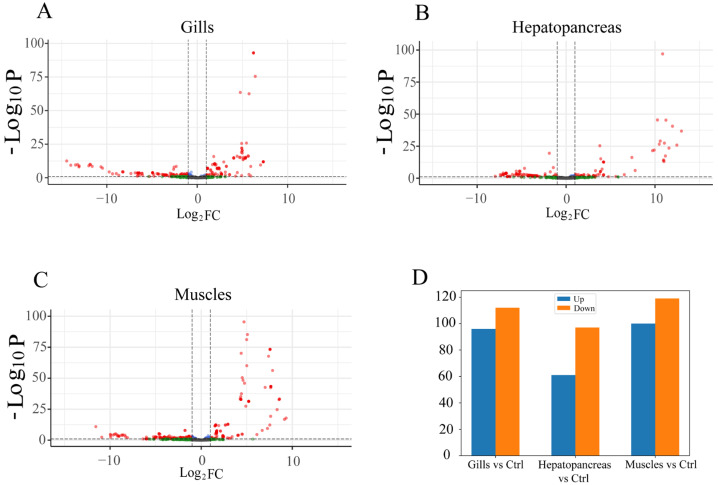
Differential expression miRNA statistics for muscle, hepatopancreas, and gill tissue. (**A**) Volcano plot of differential expression between gill and control (hepatopancreas and muscle). (**B**) Volcano plot of differential expression between hepatopancreas and control (gill and muscle). (**C**) Volcano plot of differential expression between muscle and control (gill and hepatopancreas). (**D**) Differential expression statistics for each tissue. Note: Ctrl: control; Up: upregulated miRNAs; Down: downregulated miRNAs. Note: Log_2_FC: Log_2_ fold change. This is a quantitative metric used to describe the expression level change of a biological molecule between two different conditions or time points.

**Figure 6 biology-12-01510-f006:**
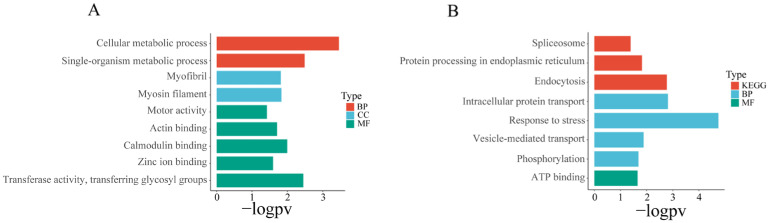
Enrichment analysis of predicted target genes of differentially expressed miRNAs in muscle and hepatopancreas. (**A**) Enrichment analysis results of predicted target genes of differentially expressed miRNAs in muscle. (**B**) Enrichment analysis results of predicted target genes of differentially expressed miRNAs in hepatopancreas. BP: biological process; MF: molecular function; CC: cellular component; KEGG: Kyoto Encyclopedia of Genes and Genomes. Gene set enrichment analysis was performed using Hiplot Pro (https://hiplot.com.cn/ (accessed on 23 May 2023)), a comprehensive web service for biomedical data analysis and visualization. In the figure below, “−logPV” is used to represent the negative logarithm (base 10) transformation of the *p*-value (probability value).

**Figure 7 biology-12-01510-f007:**
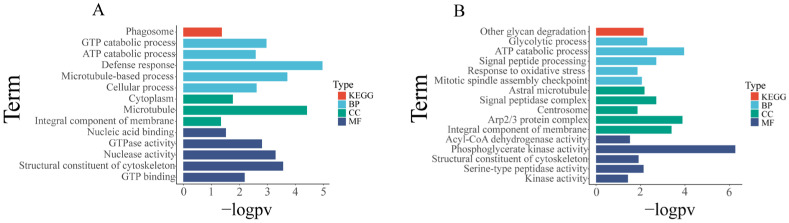
Enrichment analysis of black modules and gills. (**A**) Differential expression enrichment analysis results of the gills. (**B**) Enrichment analysis results of predicted target genes of miRNAs in the black module. BP: biological process; MF: molecular function; CC: cellular component; KEGG: Kyoto Encyclopedia of Genes and Genomes.

**Figure 8 biology-12-01510-f008:**
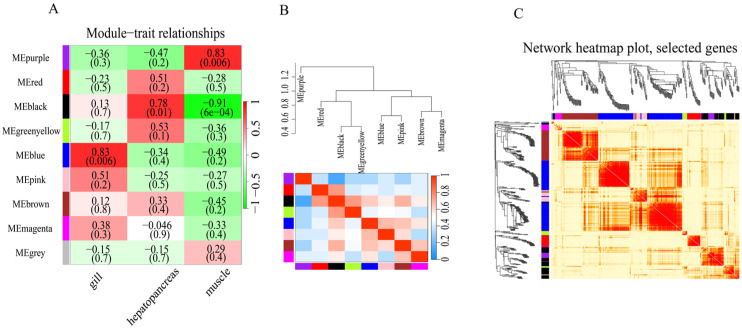
WGCNA co-expression analysis. (**A**) Module–trait relationship in each cell; the upper number indicates a correlation, and the lower number corresponds to its *p*-value. (**B**) Hierarchical clustering heatmap between modules. (**C**) Hierarchical clustering heatmap of miRNAs and modules.

**Figure 9 biology-12-01510-f009:**
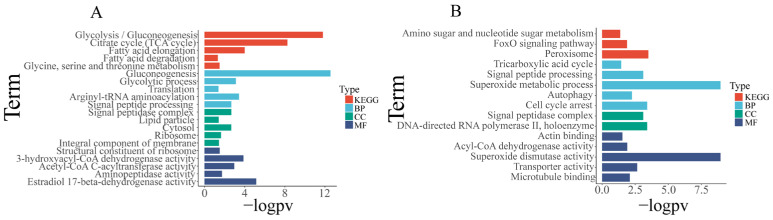
Enrichment analysis of blue and purple modules. (**A**) Enrichment analysis results of the blue module. (**B**) Enrichment analysis results of the purple module. BP: biological process; MF: molecular function; CC: cellular component; KEGG: Kyoto Encyclopedia of Genes and Genomes.

**Figure 10 biology-12-01510-f010:**
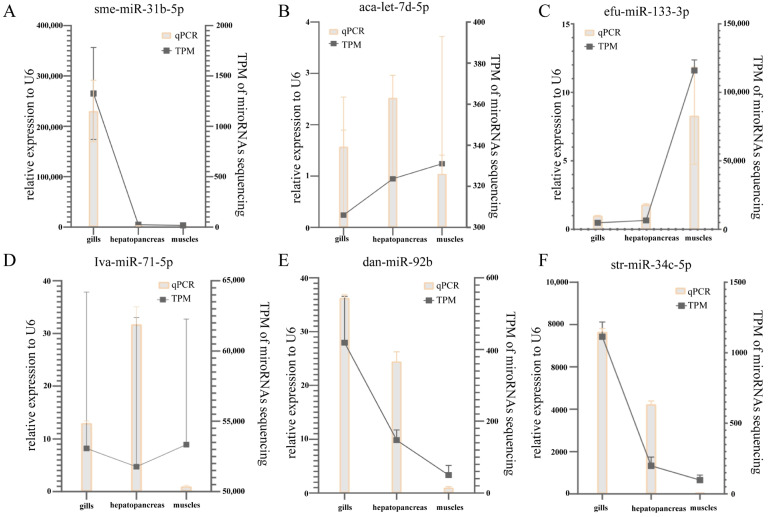
Quantitative PCR validation of miRNA expression. The expression of quantitative PCR results and TPM values (transcripts per million) of the random selected miRNAs in the gill, muscle, and hepatopancreas were are: sme-miR-31b-5p (**A**), aca-let-7d-5p (**B**), efu-miR-133-3p (**C**), lva-miR-71-5p (**D**), dan-miR-92b (**E**), and str-miR-34c-5p (**F**). The ratios to the expression level of U6 are presented on the left side of the *y*-axis. The TPM values are shown on the right side of the *y*-axis.

**Table 1 biology-12-01510-t001:** Collected specific parameters of sample specifications.

Live Weight (g)	Shell Weight (g)	Shell Length (mm)	Shell Height (mm)	Shell Width (mm)	Experimental Usage
57.80	25.50	54.05	50.50	34.71	Sequencing
49.20	23.74	51.94	49.46	30.30	Sequencing
47.40	20.22	52.11	48.57	31.16	Sequencing
38.80	17.00	50.71	47.53	27.49	qRT-PCR
49.50	21.61	54.95	51.76	30.35	qRT-PCR
47.00	22.41	52.43	48.45	30.21	qRT-PCR

**Table 2 biology-12-01510-t002:** Sequence of selected miRNAs and their forward primers for qRT-PCR.

miRNA	Sequence	Forward Primer 5′—3′
str-miR-34c-5p	TGGCAGTGTGATTAGCTGGTTG	GCAGTGTGATTAGCTGGTTGAAA
sme-miR-31b-5p	AGGCAAGATGCTGGCATAGCTGA	CAAGATGCTGGCATAGCTGAA
dan-miR-92b	AATTGCACTAGTCCCGGCCTGC	AATTGCACTAGTCCCGGCC
efu-miR-133-3p	TTTGGTCCCCTTCAACCAGCTGTA	GTCCCCTTCAACCAGCTGTAA
lva-miR-71-5p	TGAAAGACATGGGTAGTGAGATT	CGGAAAGACATGGGTAGTGAGAT
aca-let-7d-5p	AGAGGTAGTAGGTTGCATAGT	AGAGGTAGTAGGTTGCATAGTAA

**Table 3 biology-12-01510-t003:** Data output and quality distribution.

Sample Id	Clean Reads	Q30 (%)	Mapped Reads
S1	33,105,265	98.04	386,457
S2	24,612,221	98.52	138,391
S3	33,568,798	98.65	387,129
S4	20,020,543	97.71	327,220
S5	22,796,513	97.76	330,568
S6	24,537,870	97.79	480,804
S7	30,010,373	97.84	208,711
S8	33,494,607	97.92	239,902
S9	32,610,912	97.81	327,720

Note: Clean reads: the number of reads with a quality value greater than or equal to 30; Q30 (%): the proportion of bases with a quality value greater than 30. Mapped reads: clean reads compared to the reference genome.

**Table 4 biology-12-01510-t004:** Statistical results of miRNA in each sample.

Sample Id	Known-miRNAs	Novel-miRNAs	Total
S1	796	295	1091
S2	667	254	921
S3	871	316	1187
S4	540	286	826
S5	803	283	1086
S6	593	314	907
S7	602	239	841
S8	610	248	858
S9	648	266	914
Total	1047	365	1412

Note: Known-miRNA: known number of miRNAs; Novel-miRNAs: newly predicted miRNA quantities; Total: the total number of miRNAs.

**Table 5 biology-12-01510-t005:** miRNA target gene number prediction statistics.

Types	All miRNA	miRNA with Target	Target Gene
Known miRNA	1047	700	5082
Novel miRNA	365	334	3223
Total	1412	1034	7404

Note Types: miRNA type; Known miRNA: known miRNAs; Novel miRNA: newly predicted miRNAs; Total: all miRNAs; All miRNA: total number of miRNAs; miRNA with Target: predicted miRNA with predicted targets; Target gene: the number of predicted targets.

**Table 6 biology-12-01510-t006:** Statistics of miRNA target gene prediction results in different samples.

Sample Id	All miRNA	miRNA with Target	Target Gene
S1	1091	816	5694
S2	921	701	4874
S3	1187	879	6623
S4	826	638	4354
S5	1086	809	5818
S6	907	698	4767
S7	841	636	4218
S8	858	661	4372
S9	914	698	4784

Note: All miRNA: Total number of miRNAs; miRNA with Target: predicts the number of miRNAs of the target gene; Target gene: the number of predicted targets.

## Data Availability

The datasets generated for this study can be found in the National Center for Biotechnology Information (NCBI) in Bioproject: PRJNA544778, with ID: SAMN21552680 to SAMN21552688 (S1–S9).

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
