# Peer review of "microRNA-mRNA Analysis Reveals Tissue-Specific Regulation of microRNA in Mangrove Clam (Geloina erosa)"

_biology, 2023, doi:10.3390/biology12121510_

Round 1

Reviewer 1 Report

Comments and Suggestions for Authors

Liu et al performed small RNA sequencing to examine tissue-specific miRNA expression in mangrove clams. Regrettably, the manuscript lacks essential supporting data related to the three core questions they set out to address: I only see the numbers of differentially expressed miRNA were provided, but what are those miRNA?  The authors need to list them and verify them through qPCR.  I didn't notice any information on particular target genes regulated by these miRNAs, or what their physiological roles are.  None of them are fully addressed in their current manuscript. Most of the figures they provided in their manuscript are quality control (QC) related analyses.

Additionally,  The author claimed that up-regulated miRNAs could down-regulate target genes in tissues, but the data supporting this claim are absent. Additionally, the method for conducting the functional analysis outlined in section 3.5.3 is also missing from the manuscript.

Comments on the Quality of English Language

No

Author Response

Independent Review Report, Reviewer 1

Dear Reviewer:

Thank you very much for your time and effort to help us with our work. Based on your suggestion, we have fixed errors and added some content in manuscirpt. Below is a point-to-point response to your suggestion.

  1. Liu et al performed small RNA sequencing to examine tissue-specific miRNA expression in mangrove clams. Regrettably, the manuscript lacks essential supporting data related to the three core questions they set out to address: I only see the numbers of differentially expressed miRNA were provided, but what are those miRNA?  The authors need to list them and verify them through qPCR.  I didn't notice any information on particular target genes regulated by these miRNAs, or what their physiological roles are.  None of them are fully addressed in their current manuscript. Most of the figures they provided in their manuscript are quality control (QC) related analyses.

Response:

I apologize for the oversight; during the upload, we failed to include the appendix. After discussing with the editor, we have uploaded the supplementary table to the system.

All predicted miRNAs and their expression values in each tissue are presented in Supplementary Table S3. Differentially expressed miRNAs, predicted target genes, and the enrichment analysis results of these target genes can be found in Supplementary Table S1. All miRNA sequence information, including those predicted to be homologous to known miRNAs and those predicted as novel miRNAs, are available in the Supplementary Materials.

Since miRNAs in this species have not been previously reported, direct discussion of their functions poses challenges. However, miRNAs primarily function through their target genes. Hence, our functional analysis is mainly centered on the enrichment analysis of the predicted target genes (Section 3.5.3). Section 3.5.2 presents the results of differentially expressed miRNAs. Section 3.5.3 summarizes the enrichment analysis results of the predicted target genes. Section 3.5.4 consolidates the co-expression analysis, emphasizing the correlation between co-expression modules and tissues, and exploring the functions of the modules through the enrichment analysis of the predicted target genes (Supplementary Table S2). In the discussion section, we delve into the functions of tissue-specific miRNAs and find that genes regulated by miRNAs often align with the functions of the corresponding tissues, suggesting that miRNA regulation is fine-grained.

qPCR validation is crucial for expression quantification. Therefore, we selected six genes from three tissues for experimental validation. The detailed results are as follows.

We have added the following content in the Materials and Methods section of the paper:

Six miRNAs were randomly selected for validation of sequencing results through real-time fluorescence quantitative PCR. The validation experiment used clams collected from the Dongwei Mangrove in Beihai City (21.55595602°N, 109.16182882°E). Three types of tissues, muscle, gills, and hepatopancreas, were collected. Each tissue was taken from 9 individual samples, with 3 biological replicates set up. Every three tissue samples were mixed together for RNA extraction. Total RNA from each tissue was extracted using the TaKaRa MiniBEST Universal RNA Extraction Kit, following the instructions provided in the kit manual. The purity of the extracted RNA was assessed using a NanoDrop spectrophotometer, while RNA integrity was verified by agarose gel electrophoresis. Primers were designed using the Primer premier 5.0 software, as shown in Table x. The Takara Mir-X miRNA FirstStrand Synthesis and TB Green qRT-PCR Kit were used for reverse transcription of total RNA into cDNA and for subsequent qRT-PCR experiments. The primers for the internal reference gene U6 and the universal miRNA reverse primer were provided in the kit. Gene expression relative quantification experiments and analyses were carried out on the QuantStudio 3 Real-Time PCR System. The PCR reaction volume was set at 20 µL, and the PCR protocol was followed as per the product instructions. Each sample underwent 3 biological repeats and 3 technical repeats. The relative gene expression was calculated using the 2–ΔΔCt method. Data processing and plotting were performed using Graphpad 8.0.

Table 6 Sequence of selected mirnas and their primers

miRNA

Sequence

 Forward primer 5'—3'

str-miR-34c-5p

TGGCAGTGTGATTAGCTGGTTG

GCAGTGTGATTAGCTGGTTGAAA

sme-miR-31b-5p

AGGCAAGATGCTGGCATAGCTGA

CAAGATGCTGGCATAGCTGAA

dan-miR-92b

AATTGCACTAGTCCCGGCCTGC

AATTGCACTAGTCCCGGCC

efu-miR-133-3p

TTTGGTCCCCTTCAACCAGCTGTA

GTCCCCTTCAACCAGCTGTAA

lva-miR-71-5p

TGAAAGACATGGGTAGTGAGATT

CGGAAAGACATGGGTAGTGAGAT

aca-let-7d-5p

AGAGGTAGTAGGTTGCATAGT

AGAGGTAGTAGGTTGCATAGTAA

Figure 10 The relative expression levels and FPKM of selected six genes.

  1. Additionally,  The author claimed that up-regulated miRNAs could down-regulate target genes in tissues, but the data supporting this claim are absent.

Response:

Your question is very insightful. Here, we consider miRNAs that are significantly upregulated in specific tissues and whose corresponding target genes are significantly downregulated in that tissue as potential miRNA-mRNA pairs with regulatory relationships. Thus, the regulatory relationship mentioned here is of statistical significance and requires further experimental validation in the future.

  1. Additionally, the method for conducting the functional analysis outlined in section 3.5.3 is also missing from the manuscript.

Response:

Thank you. The functional analysis of differentially expressed miRNAs is a crucial part of the paper. However, the target genes discussed in this paper are based on predictive results, and thus the interpretation might carry some subjectivity. Therefore, we have placed the functional analysis from section 3.5.3 in the discussion part to incorporate our insights into the results.

Reviewer 2 Report

Comments and Suggestions for Authors

Manuscript provides valuable insights into the tissue-specific regulation of miRNAs in G. erosa, highlighting their potential roles in immune response, metabolism, and environmental adaptation Additionally, using Geloina erosa as a model organism and through miRNA sequencing and co-expression network analysis, author’s extensively studied the miRNA expression in three tissues: gills, hepatopancreas, and muscle. The results revealed newly predicted miRNAs along with known miRNAs. Functional analysis of differentially expressed miRNAs was also performed. The result showed in gills, miRNAs primarily regulated immune-related genes, substance transport, and cytoskeletal organization. Likewise, in hepatopancreas, miRNAs suppressed genes involved in shell formation and played a role in cellular motor activity and metabolism. Similarly, in muscle, miRNAs participated in metabolism and photoreceptive processes, as well as immune regulation. The data was clear, the RNA-seq analysis was done in extensive manner, and the paper was very clearly written. The work certainly adds new knowledge to the field, and in my opinion will be beneficial. Thus, I only have a few minor comments. Overall, I support publication with minor revision.

Figure 3A: X-axis: Length instead of Lengthl.

Line289: Period at the end of the sentence missing.

Section3.5.1: This section requires more description and additional explanations of the result from figure 4.

Figure 4: Sample instead of Sample (be consistent with all the figures).

Figure 4: (A) Sample correlation plot The different colors (needs period, comma, colon or semicolon).

Author Response

Independent Review Report, Reviewer 2

Dear Reviewer:

Thank you very much for your time and effort to help us with our work. Based on your suggestion, we have fixed errors and added some content in manuscirpt. Below is a point-to-point response to your suggestion.

  1. Figure 3A: X-axis: Length instead of “ Lengthl”

Response:

Thank you very much for your correction, we have corrected this error.

  1. Line289: Period at the end of the sentence missing.

Response:

Thank you very much for your correction, we have corrected this error.

  1. Section3.5.1: This section requires more description and additional explanations of the result from figure 4.

Response:

From the hierarchical clustering diagram in Figure 4A, it is found that samples of the same tissue are clustered together, indicating that the samples have good consistency. The gene expression patterns of hepatopancreas samples were specific compared to gill and muscle samples. From Figure 4B, we found that the distribution of expression density in hepatopancreas was right shift, indicating that miRNA expression values in hepatopancreas were relatively high, but the overall difference was not significant.

Sample population expression. (A) Sample Correlation Plot: Different colors in the graph represent different correlation coefficient values. Red indicates low correlation, while blue indicates high correlation. The horizontal and vertical axes represent different samples. If two samples have highly similar expression for a gene or a set of genes, their correlation coefficient will approach 1 (blue); if their expression is completely opposite, the correlation coefficient will approach -1 (red). (B) TPM stands for Transcripts Per Million: a common method used for normalizing sequencing data to represent the expression levels of genes or transcripts. Different colored curves in the chart represent different samples. The abscissa of the points on the curve represents the logarithmic value of the corresponding sample's TPM. Using logarithmic values can better differentiate between low and high expression genes since RNA sequencing data is typically skewed. The ordinate of the points represents the probability density, indicating how many genes or transcripts are at that TPM value.

  1. Figure 4: Sample instead of Sample (be consistent with all the figures.

Response:

I'm sorry, but we don't quite understand your meaning. Do you mean: Replace S1-S9 with the tissue name of the sample? We do find inconsistencies between Figure 4 and Figure 2. Figure 2 May does not distinguish between different samples of the same tissue. Therefore, we modify Figure 2 to S1-S9, and indicate the name of the sample organization in the note.

Original Figure 4:

Original Figure 2:

The revised Figure 2 and title:

Figure 2. Base quality distribution map the abscissa is the base position of the reads, and the ordinate is the average value of the base error rate of each sequencing reaction. S1-S3 representing gill, S4-S6 representing hepatopancreas, and S7-S9 representing muscle.

  1. Figure 4: (A) Sample correlation plot The different colors (needs period, comma, colon or semicolon).

Response:

Thank you very much for your correction, we have revised the original as follows. (Highlighted and red fonts represent changes)

Original text:

Figure 4. sample population expression. (A) Sample correlation plot The different colors in the plot represent different correlation coefficient values. The horizontal and vertical coordinates represent different samples. (B) The different colored curves in the TPM density distribution chart represent different samples, the abscissa of the points on the curve represents the logarithmic value of the corresponding sample TPM, and the ordinate of the points represents the probability density.

After modification:

Figure 4. sample population expression. (A) Sample correlation plot. The different colors in the plot represent different correlation coefficient values. The horizontal and vertical coordinates represent different samples. (B) Density distribution of sample expression values. The different colored curves in the TPM density distribution chart represent different samples, the abscissa of the points on the curve represents the logarithmic value of the corresponding sample TPM, and the ordinate of the points represents the probability density. S1-S3 representing gill, S4-S6 representing hepatopancreas, and S7-S9 representing muscle.

Reviewer 3 Report

Comments and Suggestions for Authors

In the manuscript “miRNA-mRNA analysis reveals tissue-specific regulation of miRNA in mangrove clam (Geloina erosa)”, Liu et al., investigated tissue-specific expression of miRNAs and their predicted targets in G. erosa. The authors identified all the miRNAs expressed in gills, hepatopancreas and muscles under physiological conditions through sequencing and in this process, identified 365 newly predicted miRNAs. The authors extended their study to identify potential targets of all the miRNAs identified and functional analysis of all these genes in three different tissues studied. Finally, the authors performed miRNA co-expression analysis to find 9 different modules which are positively corelated with one or more of the three tissues.

Overall, the authors did a good job in identifying miRNAs expressed in different tissues and building the functional network with the help of bioinformatics. I thoroughly enjoyed reading the manuscript and only have few minor concerns:

1.       The majority (if not all) of target genes are only ‘predicted’ and not ‘validated’, so these are all ‘potential’ targets and not ‘real’ targets. So, instead of ‘target genes’, I would prefer terms like ‘predicted targets’ or ‘potential targets’.

2.       Line 300:  It should be ‘Table 5’ (not Table 4).

3.       Line 324-330. Sentences are either incomplete or written vaguely. Please clearly mention how many miRNAs are differentially expressed in these tissues as compared to control. 

Author Response

Independent Review Report, Reviewer 1

Dear Reviewer:

Thank you very much for your time and effort to help us with our work. Based on your suggestion, we have fixed errors and added some content in manuscirpt. Below is a point-to-point response to your suggestion.

  1. The majority (if not all) of target genes are only ‘predicted’ and not ‘validated’, so these are all ‘potential’ targets and not ‘real’ targets. So, instead of ‘target genes’, I would prefer terms like ‘predicted targets’ or ‘potential targets’.

Response:

Thank you very much for your correction. We realize your description is more accurate. Therefore, we have revised "target genes" to "potential targets" in appropriate places in the whole paper.

  1. Line 300:  It should be ‘Table 5’ (not Table 4).

I'm sorry, this is our oversight, we have corrected this mistake.

  1. Line 324-330. Sentences are either incomplete or written vaguely. Please clearly mention how many miRNAs are differentially expressed in these tissues as compared to control.

Response:

Thank you for your advice,we have revised the original as follows.

Original text:

In comparison with the control group, 96 miRNAs were up-regulated, and 112 miRNAs were down-regulated. In the hepatopancreas versus control group comparison, 61 miRNAs were up-regulated, and 97 miRNAs were down-regulated. Furthermore, when compared to the control group, 100 miRNAs were up-regulated, and 119 miRNAs were down-regulated (Figure 5). All differential expression results are detailed in Supplementary Table S1.

After modification:

In comparison with the control group, we observed the following differential expressions of miRNAs: In the gill tissue, 96 miRNAs were up-regulated and 112 miRNAs were down-regulated; in the hepatopancreas tissue, 61 miRNAs were up-regulated and 97 miRNAs were down-regulated; in the muscle tissue, 100 miRNAs were up-regulated and 119 miRNAs were down-regulated. All differential expression results are detailed in Supplementary Table S1 (Figure 5).)
